# Diagnostic Performance of AFP, AFP-L3, or PIVKA-II for Hepatitis C Virus-Associated Hepatocellular Carcinoma: A Multicenter Analysis

**DOI:** 10.3390/jcm11175075

**Published:** 2022-08-29

**Authors:** Siyu Liu, Liyang Sun, Lanqing Yao, Hong Zhu, Yongkang Diao, Mingda Wang, Hao Xing, Wan Yee Lau, Mingcheng Guan, Timothy M. Pawlik, Feng Shen, Min Xu, Xiangmin Tong, Tian Yang

**Affiliations:** 1Department of Laboratory Medicine, Lishui Municipal Central Hospital, The Fifth Affiliated Hospital of Wenzhou Medical University, Lishui 323050, China; 2Department of Hepatobiliary Surgery, Eastern Hepatobiliary Surgery Hospital, Second Military Medical University (Navy Medical University), Shanghai 200438, China; 3Department of Medical Oncology, The First Affiliated Hospital of Soochow University, Suzhou 215006, China; 4Faculty of Medicine, The Chinese University of Hong Kong, Shatin, New Territories, Hong Kong 999077, China; 5Department of Surgery, Ohio State University, Wexner Medical Center, Columbus, OH 43210, USA; 6Department of Interventional Radiology, The Key Laboratory of Imaging Diagnosis and Minimally Invasive Interventional Research of Zhejiang Province, Zhejiang University Lishui Hospital, Lishui 323050, China; 7Cancer Center, Zhejiang Provincial People’s Hospital, People’s Hospital of Hangzhou Medical College, Hangzhou 310014, China; 8Department of Laboratory, The Fifth Affiliated Hospital of Wenzhou Medical University, Lishui 323050, China; 9The Key Laboratory of Tumor Molecular Diagnosis and Individualized Medicine of Zhejiang Province, Zhejiang Provincial People’ s Hospital (People’ s Hospital of Hangzhou Medical College), No. 158, Shangtang Road, Hangzhou 310014, China

**Keywords:** hepatocellular carcinoma, hepatitis C virus, alpha-fetoprotein, diagnosis, biomarker

## Abstract

**Background and Aim:** Alpha-fetoprotein (AFP), a lens culinaris agglutinin-reactive fraction of AFP (AFP-L3), is a protein that is induced by vitamin K deficiency or antagonist-II (PIVKA-II) that has been clinically used as a serum biomarker for early detection and diagnosis of hepatocellular carcinoma (HCC). Diagnostic performance of each serum biomarker alone, or their combinations for the detection of hepatitis C virus (HCV)-associated HCC were compared. **Methods:** Serum AFP, AFP-L3, and PIVKA-II levels were evaluated in patients with HCV-associated HCC, and those with chronic HCV infection without HCC (HCV-controls). The areas under the curve (AUC), sensitivity, and specificity were compared to identify the diagnostic performance of each serum HCC biomarker alone or in combination. **Results:** Overall, 172 HCV controls and 105 patients with HCV-associated HCC were enrolled. The AFP, AFP-L3, and PIVKA-II levels were significantly increased among patients with HCV-associated HCC when compared with HCV patients without HCC (*p* < 0.001). When these biomarkers were analyzed individually, PIVKA-II revealed the best predictive performance (AUC: PIVKA-II 0.90 vs. AFP 0.80 vs. AFP-L3 0.69, *p* < 0.001). In evaluating the combinations of any two biomarkers, the best predictive performance was found in PIVKA-II + AFP (0.93 vs. AFP + AFP-L3 0.78, *p* = 0.001; and PIVKA-II + AFP-L3 0.89, *p* < 0.001), which had no difference compared to the predictive performance of the combination of all three serum biomarkers (AFP + AFP-L3 + PIVKA-II 0.93, *p* = 0.277). Similar results were identified in the subgroups of patients with HCV-induced cirrhosis, and among patients with early-stage HCC defined by BCLC and TNM staging. **Conclusions:** The addition of the PIVKA-II test to routine AFP test maybe provide a more suitable biomarker approach to detect HCV-induced HCC in patients with HCV infection undergoing HCC surveillance.

## 1. Lay Summary

Serologic biomarkers for surveillance and early diagnosis of hepatocellular carcinoma (HCC) are an unmet need. AFP, PIVKA-II, and AFP-L3 are HCC-specific biomarkers commonly used in current clinical practice. Data from the current study demonstrated that, as the singular biomarker to detect HCV-induced HCC, PIVKA-II was revealed to have the best diagnostic performance from either AFP or AFP-L3 alone. The combination of these biomarkers improved the overall diagnostic performance, and the best predictive performance was revealed in the combination of PIVKA-II + AFP. Similar results were identified in those subgroups, regardless of cirrhosis or early-stage HCC. As such, the addition of PIVKA-II to routine AFP tests may provide a more suitable biomarker approach for the detection of HCV-induced HCC in HCV-infected patients undergoing surveillance.

## 2. Introduction

Hepatocellular carcinoma (HCC) is the most common primary liver malignancy. It is increasing in frequency worldwide, largely as a consequence of chronic liver diseases [1]. Among the etiological factors of HCC, hepatitis C virus (HCV) is a common cause. HCV-infected patients have a 1.8–8.3% annual risk and a 7–14% five-year risk of developing HCC [2,3,4]. Early diagnosis with the applicate treatment of HCC is of paramount importance to improve the survival of HCC patients. The five-year overall survival ranges from 50–80% in patients with early-stage disease and less than 10% for patients with late and advanced stages of HCC [5,6]. The benefits of HCC surveillance on survival with detection of HCC at early stages [7,8,9,10,11] have been reported resulting in many guidelines recommending HCC surveillance for at-risk populations, including patients with cirrhosis and/or chronic hepatitis infection [6,12,13,14].

Serologic biomarkers are attractive tools to assist with surveillance and early diagnosis of HCC, as well as treatment monitoring and detection of HCC [15,16]. Among known biomarkers, alpha-fetoprotein (AFP) is the most widely used clinically for the diagnosis and surveillance of HCC. However, the sensitivity and specificity of serum AFP levels are not satisfactory. Specifically, the sensitivity and specificity of AFP to detect HCC in patients with HCV have been reported to vary from 41% to 65% and from 80% to 94%, respectively [17]. Furthermore, AFP levels can be elevated in patients with nonspecific conditions like cirrhosis or chronic HCV exacerbations [17,18,19,20]. In turn, other HCC-specific serum biomarkers, including prothrombin induced by vitamin K absence-II (PIVKA-II, i.e., des-gamma-carboxy prothrombin) and lens culinaris agglutinin-reactive fraction of alpha-fetoprotein (AFP-L3) have been proposed and investigated [21,22,23]. In general, these three serum biomarkers of HCC have been variably used in clinical settings in East-Asia, as well as the West with subsequent inclusion in different HCC guidelines [24,25,26]. While the “optimal” biomarker in detecting HCC among patients with HCV is still controversial, some data suggest that overall sensitivity may be increased when all the three biomarkers are used together [16]. To this point, several studies with small sample sizes have revealed that the combination of these three serum biomarkers improved the diagnostic performance of HCC with a higher sensitivity and specificity in detecting early HCC [27,28]. Given the relative paucity of data on the accuracy of AFP when compared with AFP-L3 and PIVKA-II assays to detect HCC, the objective of the current study was to characterize the diagnostic performance of AFP, PIVKA-II, and AFP-L3 assays when each of these biomarkers was used alone, or when each biomarker was used in combination with the other biomarkers in detecting HCC among HCV-infected patients. In particular, this study aimed to define the ability of AFP, AFP-L3, and PIVKA-II assays to detect HCC among patients with early-stage HCV-induced HCC, as well as to compare with patients with HCV-related cirrhosis used as a control group.

## 3. Materials and Methods

### 3.1. Study Design and Patients

Patients with chronic HCV infection, including those patients with HCC (the HCV-HCC group) and without HCC (the HCV-control group), were retrospectively identified from the databases of four Chinese hospitals (Lishui Municipal Central Hospital, Eastern Hepatobiliary Surgery Hospital of Shanghai, the First Affiliated Hospital of Soochow University, and Zhejiang Provincial People’s Hospital) from March 2018 to May 2021. The exclusion criteria included patients with: (1) concomitant hepatitis B virus (HBV) infection; (2) obstructive jaundice; (3) specimen contamination; (4) Barcelona Clinic Liver Cancer (BCLC) stage D; (5) warfarin treatment within 1 month of the study. HCV infection was defined as HCV-RNA positivity within six months prior to surgery. The diagnosis of HCC was confirmed with ultrasound, computer tomography, or magnetic resonance imaging, and most patients had the diagnosis confirmed by histopathology based on the guidelines of the American Association for the Study of Liver Diseases [14]. Cirrhosis was diagnosed based on clinical evidence of portal hypertension or hepatic decompensation according to the aforementioned guidelines [14]. HCV-induced HCC was defined as HCC in patients with chronic HCV infection but without any evidence of alcoholic liver disease or chronic HBV infection. Two tumor staging systems were used to define the extent of the disease: BCLC staging and the 8th edition American Joint Committee on Cancer tumor-node-metastasis (TNM) staging. Early-stage HCC was defined as BCLC stage 0/A and/or 8th edition TNM stage I. Written informed consent was obtained from all the participants of this study. The study was approved by the ethics committee of each participating center, and it conformed to the ethical guidelines of the 1975 Declaration of Helsinki.

### 3.2. Measurements of Tumor Biomarkers

Peripheral blood was collected from each patient. Blood samples were separated by centrifugation at 700× *g* for 10 min. Serum was then aliquoted and immediately frozen at −80 °C until testing. The sample storage facilities were standardized at each participating hospital. Serum samples were sent to the Abbott testing centers on a regular basis. Serum concentrations of PIVKA-II and AFP were measured with the ARCHITECT immunoassay (Abbott Diagnostics, Chicago, IL, USA). Serum concentrations of AFP-L3 were detected utilizing the Fujirebio assay (Fujirebio Diagnostics, Tokyo, Japan). The lowest measurable values of AFP, PIVKA-II, and AFP-L3 were 0.6 ng/mL, 5.0 mAU/mL, and 0.5%, respectively. The technicians carrying out the laboratory tests were blinded to the diagnosis of all participants.

### 3.3. Statistical Analysis

Data count distributions were compared between groups using the χ^2^ test, with Fisher’s exact test utilized for small sample sizes. To ensure that the normality assumption was met, measurement data were compared between groups using the *t* test and the analysis of variance model on the log scale. Continuous variables as mean (standard deviation) or median (interquartile range) and categorical variables were expressed as percentages. The receiver operating characteristic (ROC) curves were used to determine the area under the curve (AUC) for AFP, PIVKA-II, or AFP-L3 alone, and for the combinations of two or three biomarkers in predicting HCC. Comparisons among AUC values were performed using the Delong test [29]. To evaluate the diagnostic performance of the combinations, a binary logistic regression was used to predict the probability of HCC. The AUC, sensitivity, specificity, positive predictive value (PPV), and negative predictive value (NPV) were used to report the diagnostic performance. A *p* value of 0.05 was considered to be statistically significant. All statistical analyses were performed using the MedCalc software 18.0 (MedCalc Software, Ostend, Belgium) and the SPSS software 25.0 (SPSS Inc., an IBM Company, Chicago, IL, USA).

## 4. Results

### 4.1. Comparisons of Clinical Characteristics between HCV-HCC and HCV Control

Of the 277 patients with HCV infection, there were 105 patients with HCC (the HCV-induced HCC group) and 172 patients without HCC (i.e., the HCV-control group). The clinical characteristics of the HCV-induced HCC group and HCV-control group are summarized in Table 1. There are significant differences in some clinical variables between the HCV-induced HCC versus the HCV-control groups including age, sex, Child-Pugh grading, and cirrhosis (all *p* < 0.05). The median levels of the three tumor biomarkers of AFP, PIVKA-II, and AFP-L3 were significantly higher in HCV-induced HCC versus HCV-control patients (87.0 versus 4.1 ng/mL for AFP, 117.0 versus 22.8 mAU/mL for PIVKA-II, and 3.5 versus 0.5% for AFP-L3; all *p* < 0.05) (Figure 1). Among patients with early-stage HCV-induced HCC, the concentrations of PIVKA-II and AFP were significantly higher in HCV-induced HCC patients than in HCV-control patients (both *p* < 0.0001, Figure 1). Of the 105 patients with HCV-HCC, 20 (19.0%) patients were diagnosed as having multiple tumors with a median largest tumor of 4.1 cm. Overall, 57 (54.3%) patients had early-stage HCV-induced HCC, based on either the BCLC 0/A or 8th TNM stage I definition.

### 4.2. Comparisons of Diagnostic Performance in Detecting HCV-HCC

The AUC, sensitivity, specificity, PPV, and NPV for AFP, PIVKA-II, and AFP-L3 alone or their combinations are presented in Table 2. Using 20 ng/mL, 10%, and 40 mAU/mL respectively as the clinical cut-off values, the performances of AFP, AFP-L3, and PIVKA-II in detecting HCC among patients with HCV-HCC were 56.2%, 36.2%, and 78.1%, respectively (Figure 2A). Of note, the performances of any two or three combinations of these three biomarkers in detecting HCC were significantly improved (range, 76.2% to 94.3% in patients with HCV-induced HCC) (Figure 2B). In particular, the AUC of PIVKA-II in diagnosing HCV-induced HCC was 0.900 (0.852–0.937), which was significantly better than the performances of AFP alone (0.802 [0.743–0.853]; *p* = 0.008) or AFP-L3 alone (0.688 [0.630–0.742]; *p* < 0.001) (Figure 2C).

To further enhance diagnostic performance, various biomarker combinations were examined. The combination of AFP + PIVKA-II had an AUC of 0.934 (0.898–0.960), which was significantly better than the other biomarker combinations (AFP + AFP-L3: AUC 0.779 [0.725–0.826], *p* < 0.001; and PIVKA-II + AFP-L3: AUC 0.895 [0.853–0.929], *p* = 0.007). However, the combination of all three biomarkers did not have a better performance to detect HCC compared with the combination of AFP + PIVKA-II (AFP + AFP-L3 + PIVKA-II: AUC 0.926 [0.888–0.954], *p* = 0.277).

### 4.3. Comparisons of Diagnostic Performance in Patients with HCV-Cirrhosis

The diagnostic performances of the biomarkers in detecting HCV-induced HCC were then examined in patients with HCV-induced cirrhosis (Appendix A). The ROC curves associated with the performances of AFP, PIVKA-II, or AFP-L3 alone and in various combinations in diagnosing HCV-induced HCC among patients with HCV-induced cirrhosis are shown in Appendix A. Similar to the findings obtained from the entire cohort, subgroup analysis of patients with HCV-induced cirrhosis demonstrated that the combination of AFP + PIVKA-II performed the best (AUC 0.931 [0.888–0.961]) over any individual or combinations of biomarkers. The combination of all three biomarkers did not perform better than the dual combination of AFP + PIVKA-II (AUC 0.928 [0.885–0.959], *p* = 0.699).

### 4.4. Comparisons of Diagnostic Performance in Detecting Early-Stage HCV-HCC

Further analyses were then performed to assess the diagnostic performances of AFP, AFP-L3, and PIVKA-II alone or in combination to predict early-stage HCV-induced HCC (Table 3). Clinical characteristics of early-stage HCC according to the BCLC and TNM staging systems are shown in Appendix A. Among patients with early-stage HCV-induced HCC (BCLC stage 0 + A, *n* = 57), PIVKA-II alone (AUC 0.897, 0.851–0.933) performed better than AFP alone (AUC 0.740, 0.678–0.795) or AFP-L3 alone (AUC 0.634, 0.568–0.697) (Figure 3). In examining the combinations of these biomarkers, the diagnosis of early-stage HCV-induced HCC (BCLC stage 0 + A) was significantly better using the combination of AFP + PIVKA-II (AUC 0.914 [0.870–0.947]) when compared with the combination of AFP + AFP-L3 (AUC 0.708 [0.644–0.766], *p* < 0.001) or the combination of PIVKA-II + AFP-L3 (AUC 0.883 [0.834–0.921], *p* = 0.095). Again, the combination of all three biomarkers failed to perform better than the combination of AFP + PIVKA-II (AFP + AFP-L3 + PIVKA-II: AUC 0.907 [0.862–0.942], *p* = 0.576). Similar results were obtained when using the 8th TNM staging system to define early-stage HCC (Figure 3). Specifically, the combination of AFP + PIVKA-II had the best performance (AUC 0.898 [0.851–0.934]) when compared with each biomarker alone or in combination of biomarkers (AUC range 0.658–0.889).

## 5. Discussion

Due to the different mechanisms of HCC in various etiology of chronic liver diseases, the diagnostic performances of AFP, AFP-L3, and PIVKA-II is probably different in HCC patients with various etiology. In the present study, we focused on patients with chronic HCV infection, considering that HCV is an important and common etiology of HCC in the West as well as in China, while China has the largest HCV-infected population in the world. Data from the current study demonstrated markedly higher levels of these biomarkers in HCV-infected patients who developed HCC than those who did not develop HCC. As a single biomarker to detect HCV-induced HCC, PIVKA-II showed the best diagnostic performance as compared to AFP or AFP-L3 alone. The combination of any of the two or three biomarkers improved overall diagnostic accuracy. Perhaps of more interest, among the different biomarker combinations, AFP + PIVKA-II showed the best diagnostic accuracy as compared to PIVKA-II + AFP-L3 or AFP + AFP-L3. However, the combination of all three AFP + PIVKA + AFP-L3 biomarkers did not further improve the accuracy than the dual combination of AFP + PIVKA-II. Similar results were obtained on subgroup analysis of patients with HCV-induced cirrhosis and/or early-staged HCV-induced HCC, regardless of whether early staging was defined by BCLC staging or TNM staging. Collectively, these data fully suggested that the combination of PIVKA-II + AFP is the most favorable biomarker combination to be used for screening or surveillance in detecting HCC among HCV-infected patients.

Current clinical guidelines recommend a liver ultrasound once every six months for surveillance of patients at high risk of developing HCC [6]. However, the sensitivity of liver ultrasound is low, and is less than 70% [7]. Furthermore, early HCC detection by liver ultrasound in patients with underlying cirrhosis has a particularly low sensitivity and a high false-negative rate. The use of liver ultrasound as an HCC surveillance tool is further limited by its operator dependency and variability. As such, serum tumor biomarkers to detect HCC may be a more convenient, cost-effective, objective, and reproducible methods to screen for HCC [12,30]. In some Asian counties, AFP, PIVKA-II, and AFP-L3 are currently used in screening and monitoring HCC under various clinical settings. HCC, however, is still difficult to diagnose in its early stages due to the relatively poor diagnostic performance of these biomarkers when used alone. As a consequence, the combined use of these biomarkers to increase the diagnostic accuracy was investigated in the current study, as different biomarkers can play complementary roles. The result of this study is that the combination of AFP + PIVKA-II biomarkers demonstrated the best performance and represented the highest diagnostic efficacy in detecting HCV-induced HCC, regardless of the presence or the absence of HCV-induced cirrhosis and the BCLC/TNM staging.

There are several reasons that explain why the diagnostic performance of the combination of AFP + PIVKA-II was comparable to the combination of all the three AFP + PIVKA-II + AFP-L3 biomarkers, and why AFP + PIVKA-II should be used as the preferred screening text for HCV-induced HCC. First, the impact on the performance of AFP and AFP-L3 is somewhat collinear, but AFP and PIVKA-II are more complementary. AFP-L3 is derived only from HCC cancer cells [31], and the concentration of AFP-L3 correlates strongly with AFP levels; thus, AFP-L3 has been suggested as a biomarker for early HCC detection due to its higher specificity than AFP [32]. A multicenter study reported AFP-L3 to have a specificity approaching 92%, yet its sensitivity was only about 37% [33]. In another multicenter study, AFP-L3 had a specificity of 97% and a sensitivity of 28% for detecting early-stage HCV-induced HCC (BCLC stage 0-A) [23]. In turn, the low sensitivity of AFP-L3 limits its potential as an HCC biomarker for screening even though its specificity is high. Second, because AFP-L3 is typically not detected when the AFP levels are <20 ng/mL, AFP-L3 cannot be used in diagnosing of HCC in individuals with an AFP < 20 ng/mL. Thus, the sensitivity of AFP-L3 is adversely affected by the AFP levels. Third, the cost of the AFP-L3 test is generally higher than that of PIVKA-II and AFP, and even higher than a combination of AFP and PIVKA-II in most medical centers. For example, at Eastern Hepatobiliary Surgery Hospital of Shanghai, where the largest number of patients were enrolled in the current study, the cost of the AFP-L3 test was 180 RMB yuan (~27.7 US dollars), while the cost of AFP and PIVKA-II were 40 RMB yuan (~6.2 US dollars) and 120 RMB yuan (~18.5 US dollars), respectively. Tests used in monitoring and screening HCC are used in a large population of patients; as such, in addition to good diagnostic performance, time spent on collection, storage, transportation, testing for repeatability/in-between laboratory variability/accuracy, and costs all need to be considered.

In the current study, more than one-half of patients with HCV-induced HCC had early-stage HCC (BCLC 0 + A or the 8th TNM stage I) at the time of diagnosis. The diagnostic performance of biomarker combinations in discriminating early-stage HCC were examined in the subgroup of patients. The combination of AFP and PIVKA-II demonstrated a good ability to discriminate early-stage HCV-induced HCC, regardless of whether early-stage HCC was defined by using BCLC staging or TNM staging. Furthermore, combining AFP-L3 with AFP and PIVKA-II did not further improve the overall diagnostic performance; rather, the dual combination of AFP + PIVKA-II performed better with a higher AUC and better sensitivity than the combination of all three biomarkers.

As we know, PIVKA-II, first described by Liebman et al. as a tumor marker for HCC, is an abnormal prothrombin that originated from an acquired defect in the post-translational carboxylation of the prothrombin precursor in HCC cells [21,22,23]. Previous studies indicated that PIVKA-II acts on many signaling pathways and played a role in HCC proliferation, invasion, and metastasis. Due to the intrinsic mechanism of PIVKA-II production, patients with vitamin K deficiency or warfarin treatment have very high serum PIVKA-II levels, which are false positives. Therefore, for patients with such a history (for example, some elderly patients who are taking warfarin with acute myocardial infarction and/or brain infarction), their serum level of PIVKA-II should be viewed with caution to avoid the possibility of misdiagnosis [21,22,23].

Several limitations should be considered when interpreting the current study. First, this retrospective study has the inherent problem of selection bias. In the future, a Phase 3 biomarker study should be carried out to definitively determine whether these biomarkers can detect preclinical HCC in patients with dysplastic nodules and chronic HCV infection in the presence/absence of liver cirrhosis. The prospective specimen and data collection in a Phase 3 biomarker study would eliminate potential biases as HCC status could be blinded to researchers on specimen and data collection. Second, the viral status of the present study population, in particular, the serum HCV-RNA levels were not known. Although no patients had received anti-HCV therapy upon study entry, the fact that the study failed to account for anti-HCV therapy could have impacted the results. Third, the diagnostic performance between ultrasound and serum biomarkers were not compared. Fourth, all patients in the present study were from China and whether the results could still be applied to other patient population needs to be externally validated to ensure generalizability. In addition, the diagnostic performance of these three commonly used biomarkers among HCC patients with other etiologies of chronic liver diseases, such as HBV infection, alcoholic liver disease, or nonalcoholic fatty liver disease, deserves to be further evaluated in the future.

In conclusion, PIVKA-II alone had a higher diagnostic accuracy than AFP or AFP-L3 alone in detecting all-stages and early-stage HCC among HCV-infected patients. Of all the combinations of AFP, PIVKA-II, and AFP-L3, the combination of AFP + PIVKA-II was the optimal biomarker panel combination for use in detecting HCC among HCV-infected patients. The data obtained from the current study should help to inform HCC screening or surveillance among HCV-infected patients.

## Figures and Tables

**Figure 1 jcm-11-05075-f001:**
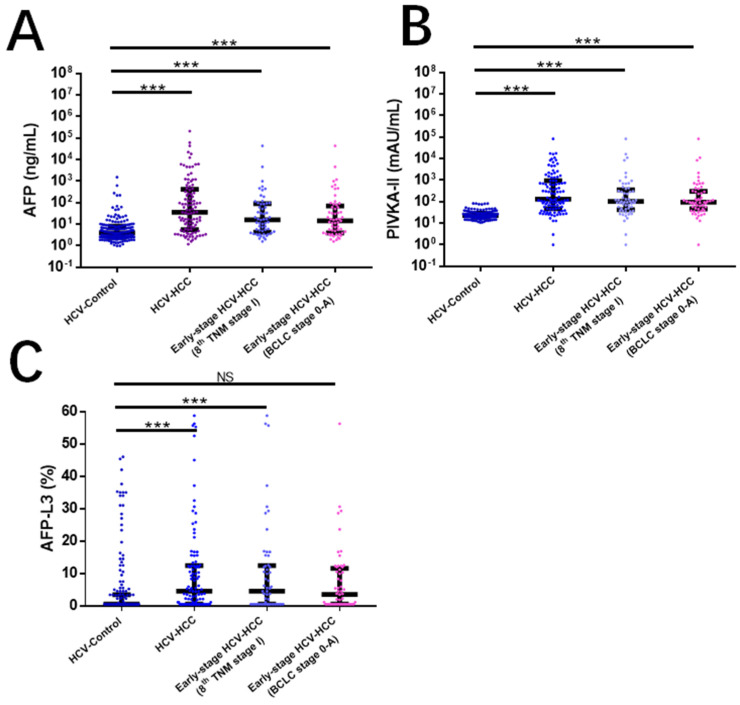
Serum concentrations of three HCC biomarkers. (**A**) AFP; (**B**) PIVKA-II; (**C**) AFP-L3. *** *p* < 0.001; NS, not significant.

**Figure 2 jcm-11-05075-f002:**
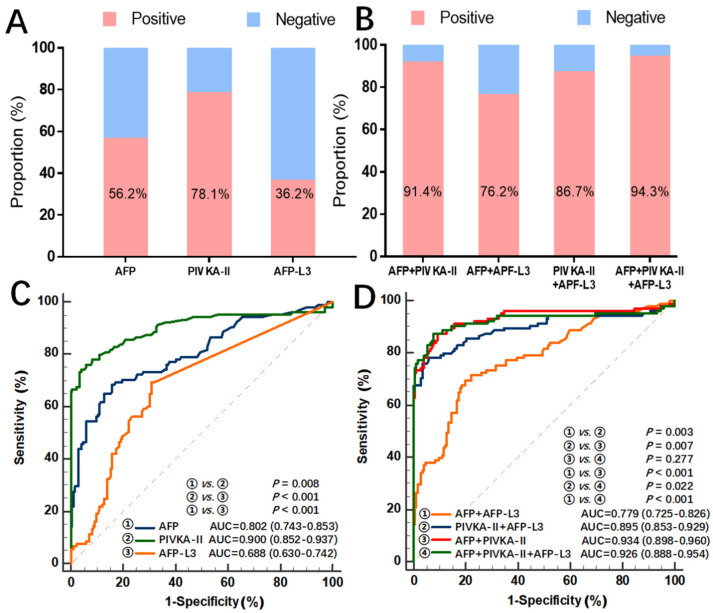
Diagnostic performance of three serum biomarkers alone and in combination for the detection of HCC among HCV-infected patients. (**A**) The positive rate of AFP, AFP-L3, and PIVKA-II alone using their respective clinical cut-off values; (**B**) The positive rate of their combinations of these biomarkers using their respective clinical cut-off values; (**C**) ROC curves of AFP, AFP-L3, or PIVKA-II alone; (**D**) ROC curves of any combination of two or three biomarkers. AUC, area under the curve; HCC, hepatocellular carcinoma; HCV, hepatitis C virus; ROC, receiver operating characteristic.

**Figure 3 jcm-11-05075-f003:**
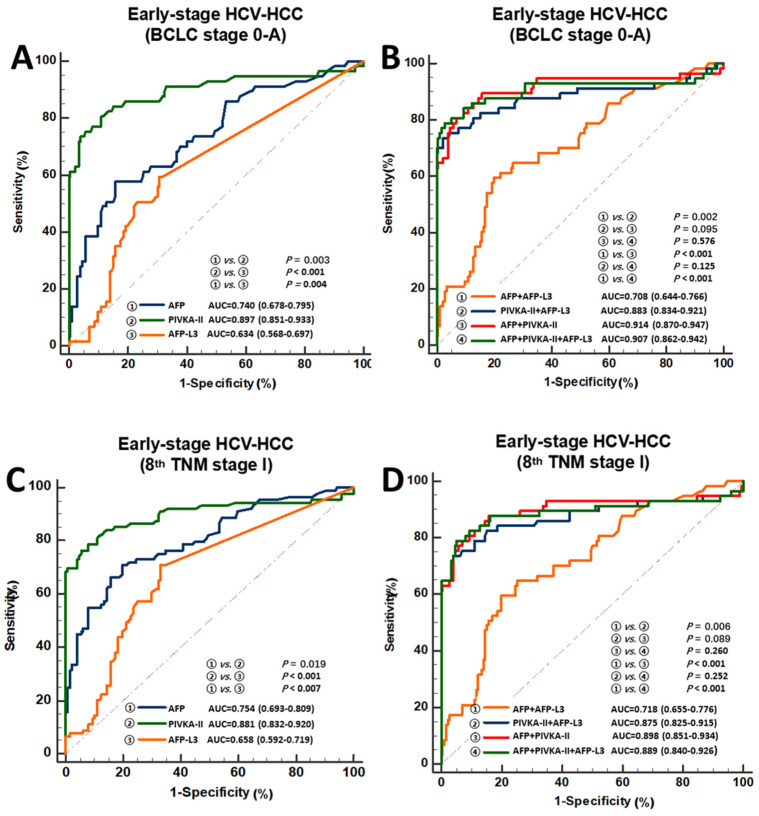
Diagnostic performance of AFP, AFP-L3, or PIVKA-II alone and in combination for the detection of early-stage HCV-induced HCC. BCLC stage 0-A: (**A**) AFP, PIVKA-II, or AFP-L3 alone; (**B**) Any combination of two or three biomarkers, 8th TNM stage I: (**C**) AFP, PIVKA-II, or AFP-L3 alone; (**D**) Any combination of two or three biomarkers. AUC, the area under the curve; HCC, hepatocellular carcinoma; HCV, hepatitis C virus.

**Table 1 jcm-11-05075-t001:** Comparisons of clinical characteristics between the HCV-induced HCC and HCV-control groups.

*n* (%)	HCV-HCC (*n* = 105)	HCV Control (*n* = 172)	*p* Value
Baseline characteristics			
Age, years *	60.8 ± 9.3	55.8 ± 12.2	<0.001
Male sex	82 (78.1)	98 (57.0)	<0.001
Child-Pugh classification			
A	92 (87.6)	161 (93.6)	<0.001
B/C	13 (11.4)	11 (7.4)	
Cirrhosis	89 (84.8)	127 (73.8)	0.033
Platelet, ×10^9^/L *	167.4 (89, 233)	148.8 (71, 187)	0.354
Bilirubin, μmol/L *	14.5 (11.5, 24.8)	14.8 (11.9, 20.9)	0.867
Albumin, g/L *	43.5 ± 5.7	45.3 ± 5.3	0.128
AFP, ng/mL *	87.0 (14.6, 1206.8)	4.1 (2.4, 8.1)	0.016
Negative (<20 ng/mL)	46 (43.7)	155 (90.1)	<0.001
Positive (≥20 ng/mL)	59 (56.2)	17 (9.9)	
PIVKA-II, mAU/mL *	117.0 (47.1, 932.0)	22.8 (19.0, 28.9)	0.005
Negative (<40 mAU/mL)	23 (21.9)	153 (89.0)	<0.001
Positive (≥40 mAU/mL)	82 (78.1)	19 (11.0)	
AFP-L3, % *	3.5 (0.5, 12.5)	0.5 (0.5, 5.4)	<0.001
Negative (<10%)	67 (63.8)	146 (84.9)	<0.001
Positive (≥10%)	38 (36.2)	26 (15.1)	
Tumor characteristics			
Tumor size, cm *	4.1 (2.6, 7.0)		
≥3 cm	72 (68.6)		
Multiple tumors	20 (19.0)		
Macrovascular invasion	15 (14.3)		
Extrahepatic metastasis	9 (8.6)		
BCLC staging system			
0/A (Early stage)	57 (54.3)		
B/C (Intermediate/advanced stage)	48 (45.7)		
8th TNM staging system			
I (Early stage)	57 (54.3)		
II/III (Intermediate/advanced stage)	48 (45.7)		

* Values are mean ± standard deviation or median with interquartile range. AFP, alpha-fetoprotein; AFP-L3, lens culinaris agglutinin A-reactive fraction of alpha-fetoprotein; HCC, hepatocellular carcinoma; BCLC, Barcelona Clinic Liver Cancer; HCV, hepatitis C virus; PIVKA-II, protein induced by vitamin K absence or antagonist-II; TNM, tumor-node-metastasis.

**Table 2 jcm-11-05075-t002:** Three serum biomarkers for detection of HCV-HCC.

	AUC (95% CI)	Clinical Cut-Off Value	Sensitivity (%) (95% CI)	Specificity (%) (95% CI)	PPV (%) (95% CI)	NPV (%) (95% CI)	Positive LR	Negative LR
AFP	0.802 (0.751–0.847)	20 ng/mL	56.2 (46.2–65.9)	90.1 (84.1–94.1)	77.6 (68.2–84.9)	77.1 (73.0–80.8)	5.7	0.5
PIVKA-II	0.903 (0.862–0.935)	40 mAU/mL	78.1 (69.0–85.6)	89.0 (83.3–93.2)	81.2 (73.6–87.0)	86.9 (82.2–90.6)	7.1	0.3
AFP-L3	0.688 (0.630–0.742)	10%	36.2 (27.0–46.1)	84.9 (78.6–89.9)	59.4 (48.6–69.3)	68.5 (65.1–71.8)	2.4	0.8
AFP + PIVKA-II	0.934 (0.898–0.960)		87.6 (79.3–93.2)	90.7 (85.3–94.6)	85.2 (78.2–90.2)	92.3 (87.8–95.2)	9.4	0.2
PIVKA-II+ AFP-L3	0.895 (0.853–0.929)		76.2 (66.9–84.0)	96.5 (92.6–98.7)	93.0 (85.8–96.7)	86.9 (82.5–90.3)	21.8	0.3
AFP+ AFP-L3	0.779 (0.725–0.826)		69.5 (59.8–78.1)	80.2 (73.5–85.9)	68.2 (60.8–74.9)	81.2 (76.2–85.3)	3.5	0.4
AFP + PIVKA-II + AFP-L3	0.926 (0.888–0.954)		87.6 (79.8–93.2)	92.4 (87.4–95.9)	87.6 (80.7–92.3)	92.4 (88.0–95.3)	11.6	0.1

AFP, alpha-fetoprotein; AFP-L3, lens culinaris agglutinin A-reactive fraction of alpha-fetoprotein; AUC, area under curve; HCC, hepatocellular carcinoma; HCV, hepatitis C virus; LR, likelihood ratio; NPV, negative prediction value; PIVKA-II, protein induced by vitamin K absence or antagonist-II; PPV, positive prediction value.

**Table 3 jcm-11-05075-t003:** Three serum biomarkers for the detection of early-stage HCV-induced HCC.

	AUC (95% CI)	Clinical Cut-Off Value	Sensitivity (%) (95% CI)	Specificity (%) (95% CI)	PPV (%) (95% CI)	NPV (%) (95% CI)	Positive LR	Negative LR
BCLC Early-stage HCV-HCC								
AFP	0.740 (0.678–0.795)	20 ng/mL	42.1 (29.1–55.9)	90.1 (84.6–94.1)	58.5 (45.0–70.9)	82.4 (78.9–85.5)	4.3	0.6
PIVKA-II	0.897 (0.851–0.933)	40 mAU/mL	77.2 (64.2–87.3)	91.9 (86.7–95.5)	69.8 (59.7–78.4)	92.2 (87.9–95.0)	9.5	0.3
AFP-L3	0.634 (0.568–0.697)	10%	31.6 (19.9–45.2)	84.9 (78.6–89.9)	40.9 (29.1–53.8)	78.9 (75.6–81.9)	2.1	0.8
AFP + PIVKA-II	0.914 (0.870–0.947)		89.5 (78.5–96.0)	84.3 (78.0–89.4)	65.4 (56.9–73.0)	96.0 (91.9–98.1)	5.7	0.1
PIVKA-II + AFP-L3	0.883 (0.834–0.921)		73.7 (60.3- 84.5)	97.7 (94.2–99.4)	91.3 (79.7–96.6)	91.8 (87.9–94.5)	31.7	0.3
AFP + AFP-L3	0.708 (0.644–0.766)		59.7 (45.8–72.4)	80.2 (73.5–85.9)	50.0 (40.9–59.1)	85.7 (81.3–89.2)	3.0	0.5
AFP + PIVKA-II + AFP-L3	0.907 (0.862–0.942)		79.0 (66.1–88.6)	97.1 (93.3–99.0)	90.0 (79.0–95.6)	93.3 (89.4–95.8)	27.2	0.2
8th TNM Early-stage HCV-HCC								
AFP	0.754 (0.693–0.809)	20 ng/mL	43.9 (30.7–57.6)	90.5 (85.6–95.1)	59.5 (46.2–71.6)	82.9 (79.3–86.0)	4.4	0.6
PIVKA-II	0.881 (0.832–0.920)	40 mAU/mL	75.4 (62.2–85.9)	91.9 (86.7–95.5)	69.4 (59.1–78.0)	91.6 (87.4–94.5)	9.3	0.3
AFP-L3	0.658 (0.592–0.719)	10%	36.8 (24.4–50.7)	84.9 (78.6–89.9)	44.7 (33.1–56.9)	80.2 (76.7–83.3)	2.6	0.7
AFP + PIVKA-II	0.898 (0.851–0.934)		87.7 (76.3–94.9)	84.3 (78.0–89.4)	64.9 (56.4–72.6)	95.4 (91.2–97.7)	6.6	0.2
PIVKA-II+ AFP-L3	0.875 (0.825–0.915)		73.7 (60.3–84.5)	95.9 (91.8–98.3)	85.7 (74.1–92.6)	91.7 (87.7–94.4)	18.1	0.3
AFP+ AFP-L3	0.718 (0.655–0.776)		64.9 (51.1–77.1)	75.0 (67.8–81.3)	46.2 (38.4–54.3)	86.6 (81.8–90.3)	3.02	0.5
AFP + PIVKA-II + AFP-L3	0.889 (0.840–0.926)		79.0 (66.1–88.6)	94.8 (90.3–97.6)	83.3 (72.3–90.5)	93.1 (89.1–95.7)	15.1	0.2

AUC, area under curve; BCLC, Barcelona Clinic Liver Cancer; HCC, hepatocellular carcinoma; HCV, hepatitis C virus; LR, likelihood ratio; NPV, negative predictive value; PPV, positive predictive value; TNM, tumor node metastasis.

## Data Availability

The data and materials used to support the findings of this study are available from the corresponding author upon request.

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
