# Peer review of "Diagnostic Performance of AFP, AFP-L3, or PIVKA-II for Hepatitis C Virus-Associated Hepatocellular Carcinoma: A Multicenter Analysis"

_jcm, 2022, doi:10.3390/jcm11175075_

Round 1
Reviewer 1 Report
Major concerns
1. The study is retrospective and the method of HCC detection in HCV patients ranges from ultrasound and CT to histopathology, as the same methods were used in patients in the group of patients without HCC we do not know how many patients were false positive and false negative for HCC .
2. The results section could be shortened and some subgroup analyzes could be transferred to supplementary material. A table of key features such as tumor size and other clinical features should be provided for the 'early HCC' group.
3. The first part of the Discussion section is mainly reapting the results
Author Response
1. The study is retrospective and the method of HCC detection in HCV patients ranges from ultrasound and CT to histopathology, as the same methods were used in patients in the group of patients without HCC we do not know how many patients were false positive and false negative for HCC.
Response 1:
Thanks for the Reviewer #1’s comment. As we have mentioned, in the present study, “the diagnosis of HCC was confirmed with ultrasound, computed tomography, or magnetic resonance imaging, and most patients had the diagnosis confirmed by histopathology based on the guidelines of the American Association for the Study of Liver Diseases (AASLD).” In other words, the gold standard for diagnosing HCC is based on the definition of AASLD. Actually, there was no comparison between histological diagnosis and imaging diagnosis involved here, so no patients were false positives and false negatives. Our study’s method was consistent with the vast majority of previously published diagnostic studies on HCC. We humbly ask Reviewer #1 if there was some misunderstanding about this. Thank you very much!
2. The results section could be shortened and some subgroup analyzes could be transferred to supplementary material. A table of key features such as tumor size and other clinical features could be provided for the 'early HCC' group.
Response 2:
Thanks for the Reviewer #1’s good suggestion. We have transferred some subgroup analyses to supplement material. As for subgroup analysis for early-stage HCC, we have adopted two different definitions for “early-stage HCC” (BCLC stage 0/A and 8th TNM stage I). Clinical characteristics of early-stage HCC according to the BCLC and TNM staging systems are shown in Supplement Table 2 of the revised manuscript (Line 473~478). Thanks again.
3. The first part of the Discussion section is mainly repeating the results.
Response 3:
Many thanks for the Reviewer #1’s comment. Some duplicate statements have been removed in the revised manuscript. Thank you very much!
Reviewer 2 Report
Authors revealed in this retrospective study that PIVKA-II was a higher diagnostic accuracy than other 2 biomarkers in detecting all-stages and early-stage HCC among HCV-infected patients. The combination of AFP+PIVKA-II was the optimal biomarker panel combination to be use in detecting HCC among HCV-infected patients.
I think some revises are needed in this manuscript.
1) We have already used both biomarkers of AFP and PIVKA-II in pre- and postoperative HCC surveillance. I think that it does not have any new evidence in this study. Please add to the comments about it in discussion.
2) There is a difficult problem in evaluating PIVKA-II in patients taking warfarin. Authors described that “warfarin treatment within 1 month of the study” in M and M section, I guess that it is uncommon situation. So, we have been so many elder patients need to take warfarin with AMI and/or brain infarction before detected HCC. How does the authors think about that situation? It should be discussion this concern.
3) Why did authors select only HCV patients and Is there any difference between HBV or NASH patients?
Author Response
Authors revealed in this retrospective study that PIVKA-II was a higher diagnostic accuracy than other 2 biomarkers in detecting all-stages and early-stage HCC among HCV-infected patients. The combination of AFP+PIVKA-II was the optimal biomarker panel combination to be use in detecting HCC among HCV-infected patients. I think some revises are needed in this manuscript.
1. We have already used both biomarkers of AFP and PIVKA-II in pre- and postoperative HCC surveillance. I think that it does not have any new evidence in this study. Please add to the comments about it in discussion.
Response 1:
Thanks for Reviewer #2’s comment. Due to the different mechanisms of HCC in various etiology of chronic liver diseases, the diagnostic performances of these three commonly-used biomarkers is probably different in HCC patients with various etiology. In the present study, we focused on patients with chronic HCV infection, considering that HCV is an important and common etiology of HCC in the West as well as in China, while China has the largest number of HCV-infected population in the world. These comments have been added into the discussion section of the revised manuscript (Line 279-284). Thank you very much!
2. There is a difficult problem in evaluating PIVKA-II in patients taking warfarin. Authors described that “warfarin treatment within 1 month of the study” in M and M section, I guess that it is uncommon situation. So, we have been so many elder patients need to take warfarin with AMI and/or brain infarction before detected HCC. How does the authors think about that situation? It should be discussion this concern.
Response 2:
Thanks to Reviewer #2 for this important comment. As we know, PIVKA-II, first described by Liebman et al. as a tumor marker for HCC, is an abnormal prothrombin originated from an acquired defect in the post-translational carboxylation of the prothrombin precursor in HCC cells. Previous studies indicated that PIVKA-II acts on many signaling pathways and played a role in HCC proliferation, invasion and metastasis. Due to the intrinsic mechanism of PIVKA-II production, patients with vitamin K deficiency or warfarin treatment have very high serum PIVKA-II levels, which are false positives. Therefore, for patients with such a history (for example, some elderly patients who are taking warfarin with acute myocardial infarction and/or brain infarction), their serum level of PIVKA-II should be viewed with caution to avoid the possibility of misdiagnosis. This comment has been added into the discussion section of the revised manuscript (Line 350-359). Thank you very much!
3. Why did authors select only HCV patients and Is there any difference between HBV or NASH patients?
Response 3:
Thanks again. As we have mentioned before, the present study focused on patients with chronic HCV infection, considering that HCV is an important and common etiology of HCC in the West as well as in China, while China has the largest number of HCV-infected population in the world. Meanwhile, the diagnostic performance of these three commonly-used biomarkers among HCC patients with other etiologies of chronic liver diseases, such as HBV infection, alcoholic liver disease, or nonalcoholic fatty liver disease, deserves to further evaluate in the future. We have added this comment into the limitation section of the revised manuscript (Line 373-376). Thanks a lot!
Round 2
Reviewer 2 Report
It is well revise. I have no problem with accept this manacript.